# The Chromosome-Level Genome of *Elaeagnus moorcroftii* Wall., an Economically and Ecologically Important Tree Species in Drylands

**Xinxing Fu [1], Jingjing Wu [2], Xiaohui Ma [2], Kunpeng Li [3,4], Hui Zhang [1], Shengdan Wu [2,\*] and Kun Sun [1,\*]**

[1] College of Life Sciences, Northwest Normal University, Lanzhou 730070, China; fuxinxing@nwnu.edu.cn (X.F.); zhanghui@nwnu.edu.cn (H.Z.)
[2] State Key Laboratory of Grassland Agro-Ecosystems, College of Ecology, Lanzhou University, Lanzhou 730000, China; wujj21@lzu.edu.cn (J.W.); maxh20@lzu.edu.cn (X.M.)
[3] State Key Laboratory of Systematic and Evolutionary Botany, Institute of Botany, Chinese Academy of Sciences, Beijing 100093, China; likunpeng@ibcas.ac.cn
[4] University of Chinese Academy of Sciences, Beijing 100049, China
\* Correspondence: wusd@lzu.edu.cn (S.W.); kunsun@nwnu.edu.cn (K.S.)

**Abstract:** *Elaeagnus moorcroftii* Wall. (Elaeagnaceae) is an important tree species naturally growing in arid Northwest China that has great economic and ecological values in drylands. In this study, we de novo assembled a chromosome-level genome for *E. moorcroftii* by using PacBio's high-fidelity (HiFi) sequencing and Hi-C-assisted assembly technology. The assembled genome size was 529.56 Mb, of which 94.56% was anchored to 14 pseudochromosomes with a contig N50 up to 28.21 Mb. In total, 29,243 protein-coding genes were annotated, and 98.5% of the Benchmarking Universal Single-Copy Orthologs (BUSCOs) were captured in the genome. Evolutionary genomic analysis showed that *E. moorcroftii* split with *Elaeagnus mollis* 9.38 million years ago (Ma), and contrasted evolutionary trajectories of gene family expansion and contraction were observed for these two closely related species. Furthermore, we identified two successive whole genome duplication (WGD) events occurred in the genome of *E. moorcroftii*, in addition to the ancient *gamma* hexaploidization event shared by core eudicots. Together, the chromosome-level genome assembly for *E. moorcroftii* decoded here provides valuable genomic information for the further genetic improvement and molecular breeding of this indigenous species in drylands.

**Keywords:** *Elaeagnus moorcroftii* Wall.; PacBio's high-fidelity sequencing; Hi-C-assisted assembly; whole genome duplication; xerophyte; drylands

## 1. Introduction

Land degradation and desertification constitute one of the most serious environmental problems facing the world. Drylands cover about 41% of the global land area and are home to more than 38% of the world's population [1,2]. Ecosystems in drylands are fragile and vulnerable to climate change and human activities [3]. The degree of desertification in these drylands is likely to increase rapidly, the areas of drylands will continue to expand, and the risk of ecological degradation will be further exacerbated [2]. Desert plants play an important role in maintaining the stability of dryland ecosystems and provide ecological services for the production and life of people in drylands. Therefore, it is very meaningful to conduct scientific research on desert plants. With the rapid development of sequencing technologies, the genomes of some important desert plants have been deciphered recently, such as the sea buckthorn (*Hippophae rhamnoides*) with medicinal and edible value [4,5], wild and perennial legume forage *Medicago ruthenica* [6], and xerophytic plant *Haloxylon ammodendron* [7].

*E. moorcroftii* is a kind of deciduous tree of the Elaeagnaceae family, up to 10 m tall, mainly distributed in the desert areas of Northwest China, including the Xinjiang, Gansu,

and Inner Mongolia provinces. It has excellent characteristics of drought resistance and salt tolerance and is an important tree species for windbreaks, sand fixation, and soil and water conservation in Northwest China [8]. This species is a non-leguminous nitrogen-fixing plant, its roots are symbiotic with *Frankia* actinomycetes, which can play the role of biological nitrogen fixation and soil improvement [4,9], and its rhizosphere arbuscular mycorrhizal fungi can improve their resistance to salt stress [10]. Therefore, *E. moorcroftii* can be introduced to barren desert and saline-alkali land for soil improvement and afforestation, which has important ecological value. In addition, this species has high economic value because of its edible fruits, medicinal whole plants, and ornamental flowers. The species exhibits a high fruit yield; the fruit can be eaten directly and used in jam, vinegar, wine, pastries, livestock feed, etc. [11]. The branches, leaves, and flowers of the species have remarkable observed biological activities and are widely used to treat many health issues like aging, burns, dyspepsia, diarrheal, pain, bronchitis, and neurasthenia [11,12]. The flowers are attractive and aromatic and are also used for extracting aromatic oil [11]. However, the molecular-level study of *E. moorcroftii* is limited to taxonomic relationships [8]. The lack of genomic information hinders a comprehensive understanding of the evolutionary history and unique biology of *E. moorcroftii*.

In this study, we first report a chromosomal-level genome assembly of *E. moorcroftii* (2n = 2x = 28) with PacBio's long-read single molecule high-fidelity (HiFi) reads and high-throughput chromosome conformation capture (Hi-C) data, and then we used it to explore the evolutionary trajectories of *E. moorcroftii* and other Elaeagnaceae species, including *H. rhamnoides* [4,5] and *E. mollis* [13] as recently published, by comparative genomic analysis. The genome sequence of *E. moorcroftii* presented here will provide valuable genomic resources for further in-depth study and utilization of this indigenous species in drylands.

## 2. Materials and Methods

### 2.1. Plant Materials and Sequencing

The fresh young leaves used for genomic DNA sequencing were collected from an adult plant of *E. moorcroftii* growing in Minqin Desert Botanical Garden, Gansu Province, China. The total genomic DNA was extracted using a modified trimethylammonium bromide (CTAB) method [14]. For PacBio's HiFi sequencing, SMRTbell libraries were constructed using the protocol of Pacific Biosciences with 20 kb inserts and sequenced using circular consensus sequencing (CCS) mode on a PacBio Sequel II platform, generating a total of more than 28 Gb high-quality CCS reads. For Hi-C sequencing, cross-linked chromatin was first digested with *Dpn* II, end-labeled with biotin-14-dATP, and then used for in situ DNA ligation. The ligated DNA was sheared into 300–600 bp fragments, blunt-end repaired, purified through biotin–streptavidin-mediated pull-down, and then sequenced on the Illumina HiSeq 2500 platform, generating a total of more than 52 Gb raw sequencing data.

### 2.2. Genome Assembly and Assessment

The high-quality HiFi reads were assembled into contigs using hifiasm v0.14 [15] with default parameters. The Hi-C data were aligned to the contig assembly using JUICER v.1.5 [16]. The contigs of the *E. moorcroftii* assembly were further clustered, ordered, and oriented onto chromosomes based on the contact frequency calculated from the mapped Hi–C read pairs by 3d-DNA pipeline [17] with parameters of '-m haploid –r 2'. Orientation and placement errors were manually corrected via Juicebox Assembly Tools (https://github.com/aidenlab/Juicebox (accessed on 17 November 2021)).

The completeness of the genome assembly of *E. moorcroftii* was assessed by transcript alignment and Benchmarking Universal Single-Copy Orthologs (BUSCO) analysis [18]. RNA-seq reads were assembled and mapped to the genome by HISAT2 v2.1.0 [19]. BUSCO analysis of the final assembly and annotation was performed using BUSCO v4 [18] with the Embryophyta obd10 database to evaluate the completeness of the reference genome of *E. moorcroftii*.

### 2.3. Genome Annotation

Tandem repeats in the *E. moorcroftii* genome were annotated using GMATA v2.2 [20] and Tandem Repeats Finder v4.09 [21]. The transposable elements (TEs) in the genome were predicted using a combination of homology-based and de novo approaches. In the homology-based approach, we used RepeatMasker v.4.1.0 [22] with the known repeat database Repbase v.21.01 [23] to search for the TEs in the *E. moorcroftii* genome. For the de novo approach, we used MITE-Hunter [24] and RepeatModeler v.2.0 [25] to construct a de novo repeat sequence database for *E. moorcroftii* and then used RepeatMasker v.4.1.0 [22] to search for repeats in the genome. After removing overlapping repeats, the repeats identified by different methods were combined into the final repeat annotation.

Protein-coding genes were predicted based on the repeat masked genome using three approaches, including homology search, de novo prediction, and estimation from transcriptome evidence. Homology-based prediction was performed using GeMoMa v1.3.1 [26] with the protein sequences of four closely related species in eudicots, *E. mollis* [13], *Ziziphus jujuba* [27], *Arabidopsis thaliana* [28], and *Vitis vinifera* [29]. The de novo prediction was conducted using Augustus v3.2.3 [30] with default parameters. RNA-seq based gene prediction was performed using STAR v2.7.3a [31], Stringtie v2.0.1 [32], and PASA v2.0.2 [33], and the public available RNA-seq data for mixed tissue samples including the leaf, root, stem, and fruit were downloaded from the National Center for Biotechnology Information (NCBI) under the accession numbers of SRR12569922, SRR12569923, and SRR12569924. Finally, the outputs from the above three approaches were integrated into a final gene set by EvidenceModeler (EVM) v1.1.1 [34].

For gene function annotation, the predicted gene models were blasted against the SwissProt v5.3 (https://www.expasy.org/resources/uniprotkb-swiss-prot (accessed on 29 July 2021)), NCBI non-redundant (NR) (https://www.ncbi.nlm.nih.gov/refseq/about/nonredundantproteins/ (accessed on 29 July 2021)), and Clusters of Orthologous Groups for Eukaryotic complete Genomes (KOG) v5.3 (http://genome.jgi-psf.org/help/kogbrowser.jsf (accessed on 29 July 2021)) databases for the best matches using BLASTP with an *E*-value cut-off of 1e-5. The protein motifs and domains were annotated using InterProScan v5.31 [35]. The Gene Ontology (GO) entries were searched using Blast2GO v2.5 [36]. Pathway information for each gene was assigned using the Kyoto Encyclopedia of Genes and Genomes (KEGG) v5.24 database [37].

In addition, the non-coding RNA genes were annotated. tRNAs were predicted by tRNAscan-SE v1.3.1 [38] with eukaryote parameters. MicroRNA, small nuclear RNA, and small nucleolar RNA were predicted using INFERNAL v1.1.2 [39] based on the Rfam [40] and miRbase databases [41]. The rRNAs and their subunits were predicted using RNAmmer v1.2 [42].

### 2.4. Comparative and Evolutionary Genomic Analysis

The protein sequences of *E. moorcroftii* and ten other sequenced plant species, *H. rhamnoides*, *E. mollis*, *Malus domestica*, *Prunus persica*, *Z. jujuba*, *Morus notabilis*, *Cannabis sativa*, *Populus trichocarpa*, *A. thaliana*, *V. vinifera*, were used for the phylogenetic analysis (see Table S1 for the summary of genomic information of these species). Single-copy orthogroups among the 11 species were identified using OrthoFinder v2.3.11 [43]. The amino acid alignments of each single-copy orthogroup were aligned by MAFFT v.7 [44], and nucleotide alignments were generated according to the corresponding amino acid alignments using PAL2NAL [45]. A maximum likelihood phylogeny was constructed based on the concatenated alignments of all single-copy genes using IQ-TREE v.1.6.12 [46]. The species divergence time was estimated using the program MCMCTREE in the PAMLv.4.9 package [47]. We selected three fossil calibration points from the TimeTree database (http://www.timetree.org (accessed on 1 March 2022)) for the split of: (1) *Malus-Prunus* at 30–61 million years ago (Ma); (2) *Morus-Cannabis* at 53–97 Ma; (3) *Populus-Arabidopsis* at 98–117 Ma. In addition, the time calibration of family Rhamnaceae (that is the split of *Ziziphus* with three Elaeagnaceae species) was set to >99 Ma based on an old fossil in the extant genus *Phylica* of Rhamnaceae [48]. The gene

family expansion and contraction of 11 species were analyzed using CAFE v3.1 [49], and the expanded and contracted gene families in *E. moorcroftii* were subjected to GO enrichment to analyze their functions. The different modes of gene duplications were identified by DupGen_finder software [50].

### 2.5. Whole Genome Duplication Analysis

The protein-coding sequences of *E. moorcroftii*, *H. rhamnoides*, and *E. mollis* were self-aligned and aligned with each other using BLASTP with an *E*-value cut-off of 1e-5. Syntenic blocks within a genome or between genomes were identified using MCScanX [51] with default parameters based on the above protein-sequence alignments. For each syntenic gene pair, the synonymous substitution rate (*Ks*) was calculated using the Nei-Gojobori method [52] by yn00 program of the PAML package [47]. The macro-syntenic relationships among the three species were visualized by the python version of MCScan software (https://github.com/tanghaibao/jcvi/wiki/MCscan-(Python-version) (accessed on 29 July 2021)).

## 3. Results

### 3.1. Genome Assembly of E. Moorcroftii

To obtain a high-quality reference genome of *E. moorcroftii*, we used a combination of HiFi long reads and Hi-C data to construct a chromosome-level assembly. We generated a total of 28 Gb high-quality HiFi long reads, with a maximum and average read length of 64.20 and 14.78 kb, respectively (Table S2 and Figure S1, distribution of PacBio's HiFi sequencing data). De novo assembly of these HiFi reads with hifiasm v0.14 [16] yielded a preliminary assembly comprising 168 contigs with contig N50 size of 28.21 Mb, and the total length of the assembled genome was 529.56 Mb (Table 1). A total of 52 Gb Hi-C reads were generated to refine the assembly, resulting in 500.73 Mb (94.56%) of the contig sequences anchored onto 14 chromosomes (Figure 1, Table 1).

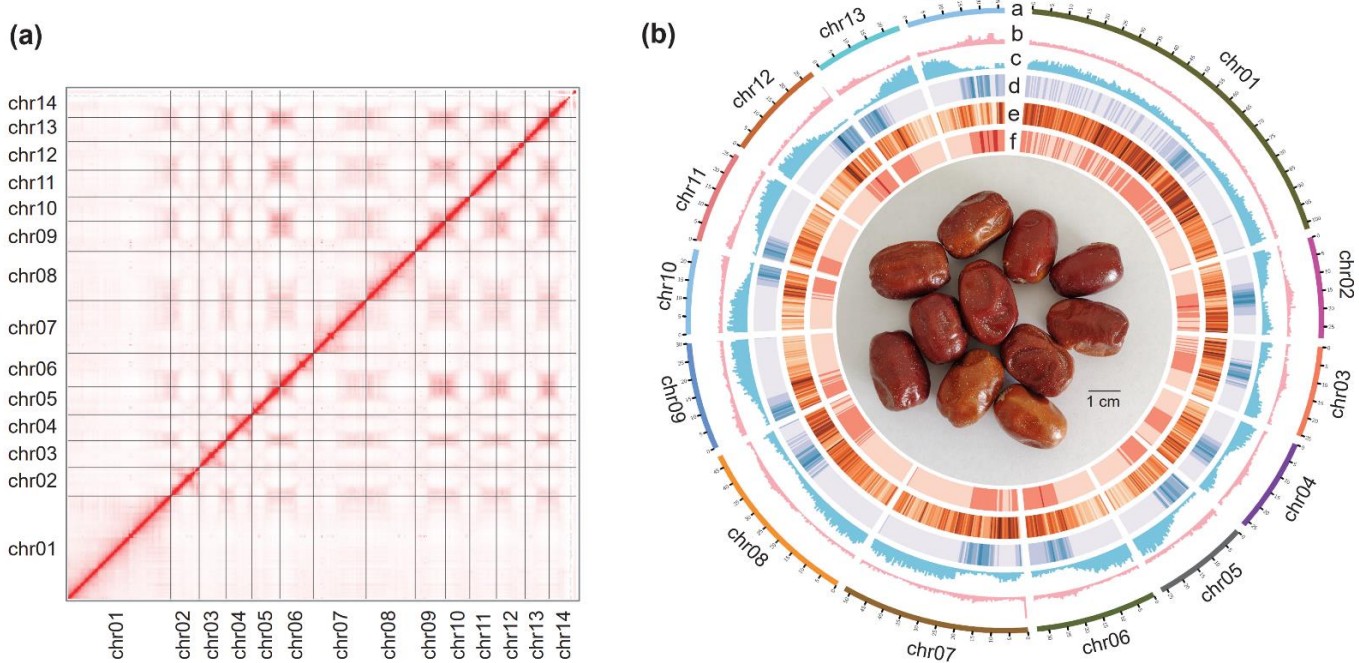

**Figure 1.** The genome features of *E. moorcroftii*. (**a**) Heatmap of Hi-C interactions for 14 pseudochromosomes. (**b**) Circos plot showing the genomic landscape of *E. moorcroftii*. The tracks from outer to inner circles indicate the following: a: 4 pseudochromosomes in megabases; b: GC content; c: gene density; d: density of *Gypsy* LTR retrotransposons; e: density of *Copia* LTR retrotransposons; f: LTR retrotransposons density. The center of the circos plot shows the fruits of *E. moorcroftii* (photo by X.F.).

**Table 1.** Assembly and annotation statistics of *E. moorcroftii* genome.

| Assembly | |
|---|---|
| Length of genome assembly (Mb) | 529.56 |
| Anchored to chromosome (Mb) | 500.73 |
| Contig N50 (Mb) | 28.21 |
| Longest contig (Mb) | 101.76 |
| BUSCO score of assembly (%) | 96.7% |
| **Annotation** | |
| GC content | 30.39% |
| Percentage of repeat sequences (%) | 60.95% |
| Number of protein-coding gene (%) | 29243 |
| Average gene length (bp) | 4318.92 |
| Average exon length (bp) | 220.21 |
| BUSCO score of annotation (%) | 98.5% |

To assess the quality and completeness of this assembled genome, we first mapped RNA-seq reads to the assembled genome, more than 93% of which were properly mapped (Table S3). Furthermore, BUSCO analysis for the genome assembly showed that 96.7% of the 1614 core plant genes was captured, including 95.8% complete BUSCOs and 0.9% fragmented BUSCOs (Table S4). These evidences together indicated that the *E. moorcroftii* assembly has high quality and completeness.

### 3.2. Annotation of the E. Moorcroftii Genome

We identified 322.78 Mb of non-redundant repetitive sequences in the *E. moorcroftii* genome, representing 60.95% of the genome assembly (Table S5). Long terminal repeat (LTR) retrotransposons were the most abundant type, accounting for 31.95% of the whole genome, of which *Copia* and *Gypsy* were the two most frequent LTR types, accounting for about 16.02% and 13.30% of the genome, respectively (Table S5).

Using genomic and transcriptomic data, we predicted 29,243 protein-coding gene models in the *E. moorcroftii* genome, with an average gene length of 4318.92 bp and an average exon length of 220.21 bp (Table 1). Among the predicted protein-coding genes, 95.10% were annotated through at least one of the following protein-related databases: the NCBI NR database (94.75%), the SwissProt protein database (81.38%), the KOG database (91.24), the InterProScan database (86.10), the GO database (64.91%), and the KEGG database (3.87%) (Table S6). In total, 95.10% of the protein-coding genes were functionally annotated by various databases. In addition, our annotation captured 98.5% of BUSCOs, including 97.4% complete gene models plus 1.1% fragmented gene models (Table S7).

We also predicted 183 micro RNAs (miRNAs), 2461 small-nuclear RNAs (snRNAs), 1636 transfer RNAs (tRNAs), and 20,568 ribosomal RNAs (rRNAs), with calculated average lengths of 124.86, 107.52, 75.21, and 238.47 bp, respectively (Table S8).

### 3.3. Evolutionary History of E. Moorcroftii

To investigate the evolutionary history of *E. moorcroftii*, we performed a gene family clustering using *E. moorcroftii* and ten other representative angiosperm species, including two related species of the Elaeagnaceae family (*E. mollis*, *H. rhamnoides*), five species of the same order Rosales (Rosaceae: *M. domestica*, *P. persica*, Rhamnaceae: *Z. jujuba*; Moraceae: *M. notabilis*; Cannabaceae: *C. sativa*), and three outgroup species of eudicot clade (*A. thaliana*, *P. trichocarpa*, and *V. vinifera*) (Table S1). We identified 175 single-copy orthogroups and used them to reconstruct the phylogenetic tree of *E. moorcroftii* and ten other plant species (Figure 2). The results showed that *E. moorcroftii* clustered a monophyletic group with its related species *E. mollis* and *H. rhamnoides*, which in turn formed sister to *Z. jujuba* of Rhamnaceae. The divergence time between *E. moorcroftii* and *E. mollis* was estimated to be around 9.38 million years ago (Ma), and the most recent common ancestor of two *Elaeagnus* species split with *H. rhamnoides* occurred at about 30.52 Ma (Figure 2).

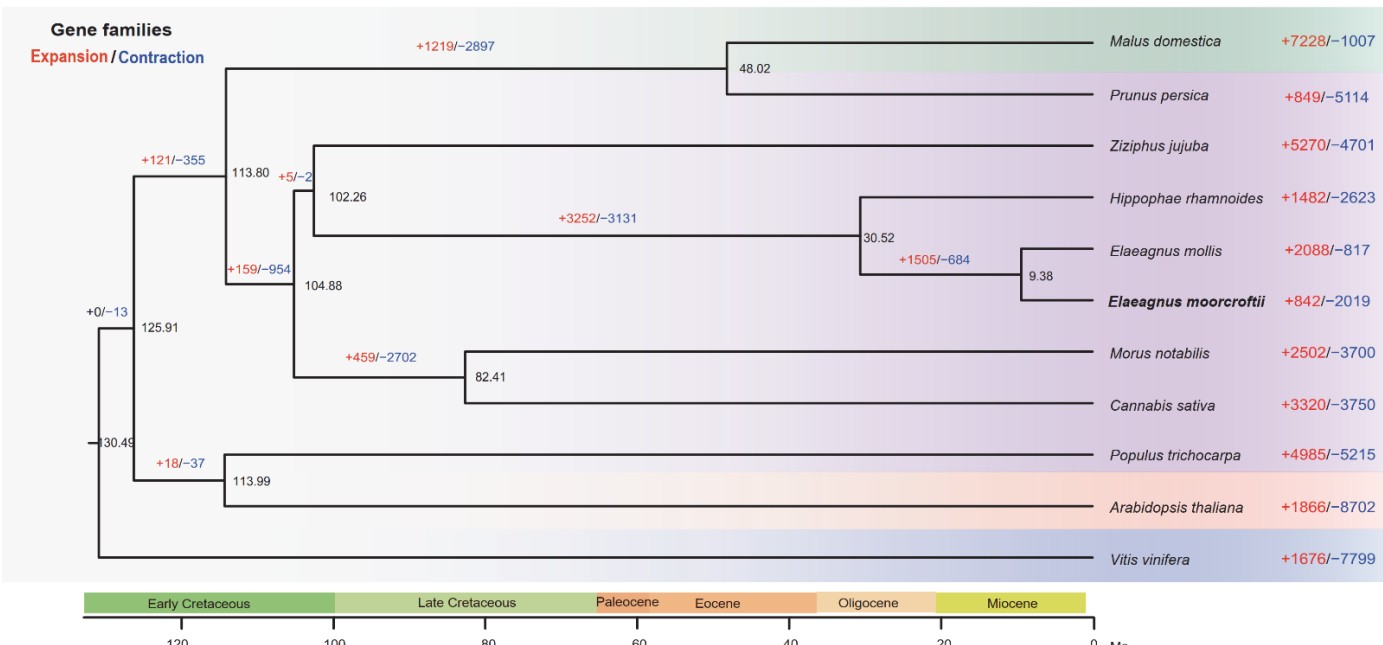

**Figure 2.** Comparative and evolutionary genomic analysis of *E. moorcroftii* and 10 other plant species. A phylogenetic tree among 11 species was reconstructed based on 175 single-copy genes, and their divergence times were also estimated. The numbers of expansion (blue) and contraction (red) gene families are shown above the branches.

Furthermore, to explore lineage-specific dynamic changes in gene families, the expansion and contraction of gene families based on the birth-and-death model were identified by CAFE v3.1 [49]. We detected 842 expansion and 2019 contraction gene families in *E. moorcroftii* genome, relative to the most recent common ancestor of *E. moorcroftii* and *E. mollis* (Figure 2), whereas an opposite trend was observed in its close relative *E. mollis*, which included more expanded (2088) than contracted (817) gene families. This difference might be partly resulted from the different levels of contribution from various modes of gene duplications. For example, dispersed duplications contributed to more in *E. mollis* than in *E. moorcroftii* with relation to the expanded gene families (16.45% versus 7.64%). In contrast, tandem duplications contributed to a higher percentage of expanded genes in *E. moorcroftii* than in *E. mollis* (14.34% versus 6.40%) (Figure S2). GO enrichment analysis revealed that the expanded gene families in *E. moorcroftii* were mainly involved in multiple biosynthetic processes (e.g., diterpenoid biosynthetic process and ATP biosynthetic process) and various metabolic process (e.g., ATP metabolic process, diterpenoid metabolic process, and purine-containing compound metabolic process), as well as immune responses (innate immune response and immune system process) (Figure S3), while contracted gene families were mainly related to protein depolymerization biological process and their enriched molecular function including ADP binding, ion channel activity, hormone binding, etc. (Figure S4).

A detailed comparative analysis among *E. moorcroftii*, *E. mollis*, *H. rhamnoides*, and *Z. jujuba* identified 12,118 common gene families shared by these four species, and 278 gene families uniquely appeared in *E. moorcroftii* (Figure S5). This number is comparable with the unique gene families in *E. mollis* (388) and *H. rhamnoides* (396), but it is much fewer than the unique gene families in *Z. jujuba* (2083) (Figure S5). The unique gene families are mainly involved in cellular respiration and oxidative phosphorylation biological processes (Figure S6).

*3.4. Whole-Genome Duplication Events in E. Moorcroftii*

The distributions of synonymous substitutions per synonymous site ($K_S$) of paralogous gene pairs in *E. moorcroftii* genome showed two recent clear peaks around 0.38 and

0.45 (Figure 3a). Similar $K_S$ peaks also were identified in the genomes of its two closely related species, *E. mollis* and *H. rhamnoides* (Figure 3a). Moreover, the large-scale gene duplications in three species occurred earlier than the time of their divergence (Figure 3a). Therefore, these two closely occurring peaks might reflect two successive whole-genome duplication (WGD) events shared by three Elaeagnaceae species. To further confirm this, an intragenomic synteny analysis of *E. moorcroftii* identified one syntenic block corresponding to three homologous syntenic blocks (Figure S7), again supporting two relatively recent WGD events that occurred in *E. moorcroftii*. In addition, syntenic analysis among the genomes of *E. moorcroftii* and the most closely related species (*E. mollis* and *H. rhamnoides*) was performed to explore WGD history that occurred in the Elaeagnaceae family. By using a *Vitis vinifera*, a basal core eudicot lineage lacking any further WGD event after the ancient *gamma* event shared by core eudicots [53], genome as a reference, the intergenomic synteny comparisons between *V. vinifera* and *E. moorcroftii* revealed a clearly syntenic depth ratios of 1:4, indicating two WGD events occurred after the split of the two species (Figure 3b). Furthermore, the genomes of *E. moorcroftii* and *E. mollis* present highly conservative synteny, and consistent syntenic depth ratios were found for *E. mollis* and *H. rhamnoides* (Figure 3b). Thus, the integrated evidence showed that two successive WGD events might have occurred in the common ancestor of *E. moorcroftii*, *E. mollis*, and *H. rhamnoides*.

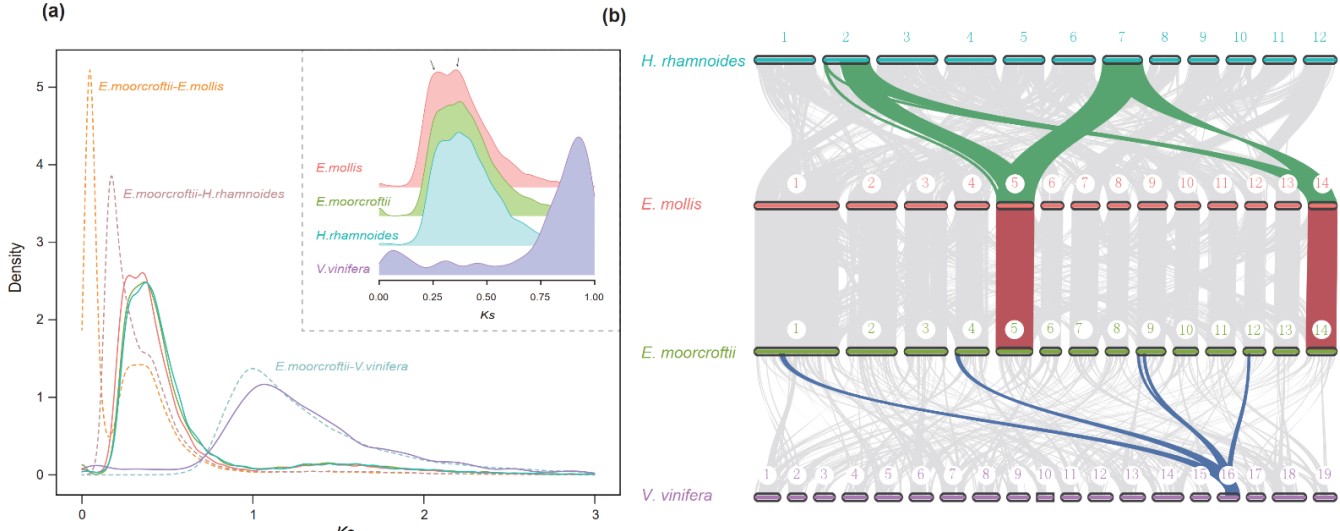

**Figure 3.** Integrating $K_S$ and synteny analyses reveal two successive WGD events in *E. moorcroftii*. (**a**) The distributions of $K_S$ of paralogous gene pairs of *E. mollis* (pink line), *E. moorcroftii* (green line), *H. rhamnoides* (blue line), and *V. vinifera* (purple line). The dashed lines represent $K_S$ distributions related to the species divergence of the corresponding species pairs. (**b**) Macro-syntenic comparisons among three Elaeagnaceae species and *V. vinifera*.

## 4. Discussion

### 4.1. A High-Quality Dryland Tree Species Genome

Land degradation and desertification is an ongoing environmental problem that threatens the sustainable development of human beings, and this is particularly serious for lives in drylands [1–3]. A native or indigenous species naturally growing in drylands, to some extent, might hold the key to solve this problem. Recently, a growing number of studies have focused on decoding the genomic information of typical arid plants, for example, *Populus euphratica* [54], *H. rhamnoides* [4,5], *H. ammodendron* [7], etc. To our knowledge, besides the well-known desert poplar *P.euphratica*, our genome assembly for *E. moorcroftii* represents the second genome for high-tree species naturally growing in arid northwest China. Meanwhile, the chromosome-level genome of *E. moorcroftii* is of high quality, with contig N50 up to approximately 30 Mb (Table 1), which is much higher than

that of its two closest relatives (*H. rhamnoides* with N50 of 3.56 Mb in [4] and 2.15 Mb in [5], and *E. mollis* with N50 of 18.40 Mb) of Elaeagnaceae with published genomes.

### 4.2. Differential Evolutionary Dynamics of Gene Families between E. Moorcroftii and E. Mollis

Our phylogenomic and molecular dating analyses showed that the divergence time between *E. moorcroftii* and its close relative *E. mollis* was about 9.38 Ma. After the split from their most recent common ancestor (MRCA), these two species experienced independent gene family evolutionary dynamics, which can be observed by the contrasted expansion and contraction trend (Figure 2). The number of contracted gene families is more than twice the expanded gene families in *E. moorcroftii*, while this trend is opposite in *E. mollis*, which have more expanded than contracted gene families (Figure 2). A detailed analysis suggested that these differential evolutionary dynamics of gene families *between E. moorcroftii* and *E. mollis* might partly result from contribution levels of different modes of gene duplications (Figure S2). Nevertheless, the opposite evolutionary trend of gene families might reflect their unique biology of these two species.

### 4.3. The Potential Evolutionary Significance of two Successive WGD Events in E. Moorcroftii

For Elaeagnaceae plants, a recent genome study of *H. rhamnoides* revealed it experienced a recent whole-genome duplication after the divergence between *H. rhamnoides* and the most closely related species jujube, with a signature *K*s peak at c. 0.38 [4]. However, almost simultaneously, another study of *H. rhamnoides* genome indicated that it experienced two rounds of WGDs, one recent WGD (*K*s peak at ~0.38) and one older WGD (*K*s peak at ~0.45) [5]. In this study, our integrated *K*s and synteny analyses provide strong evidence for the occurrence of two successive WGD events in *E. moorcroftii* genome (similar with the report by [5]), in addition, these two events possibly being shared by *E. moorcroftii* and its two close relatives (*E. mollis* and *H. rhamnoides*). It has been well-acknowledged that the frequent occurrence of WGD events has played an important role on the plant's adaptive evolution and diversification [55–59]. For example, the co-retained duplicated genes, after multiple independent WGDs in different angiosperms lineages were selected by environmental stresses during the Cretaceous–Paleocene (K-Pg) mass extinction, possibly enhanced the plant's survival and adaptation [59]. Furthermore, it was demonstrated that not only did the MRCA of angiosperms experience an ancient WGD event, but the MRCA of all extant Gymnosperms also shared an ancient WGD event, as revealed by a recent report of the *Cycas* genome, which may have contributed to seed-related morphology innovation [60]. Considering the evolutionary significance of the recognized WGD, it should be postulated that these two successive WGD events might have contributed to the evolution and adaptation of *E. moorcroftii*, although it needs to be further investigated.

### 5. Conclusions

As an indigenous species naturally growing in arid Northwest China, *E. moorcroftii* presents ideal tree species with great economic and ecological values that can be widely cultured in drylands for economic development and ecological restoration. In this study, the high-quality genome sequence of *E. moorcroftii* decoded here represents a small but essential step forward to understanding the evolution and adaptation of this species in drylands. Our results not only provided new insights into the genomic evolution of Elaeagnaceae species but also revealed clear evidence for two successive WGD events that occurred in the common ancestor of *E. moorcroftii*, *E. mollis*, and *H. rhamnoides*. Further, by using the *E. moorcroftii* genome as reference, more studies should be encouraged to investigate the unique biology of this species. For example, how could this species respond to long-term drought and salt stresses? Moreover, whole-genome resequencing analysis of samples with certain properties (such as resistance to abiotic factors) is helpful to screen for better germplasm resources and breed new varieties. In total, the genomic resource of *E. moorcroftii* paves ways for the further genetic improvement and widespread cultivation of this indigenous species in drylands.

**Supplementary Materials:** The following supporting information can be downloaded at: https://www.mdpi.com/article/10.3390/d14060468/s1, Figure S1: The distribution frequency of PacBio's HiFi sequencing data; Figure S2: The different modes of gene duplications related to expanded and contracted gene families in *E. moorcroftii* and *E. mollis*; Figure S3: The enriched GO terms for the expanded gene families of *E. moorcroftii*; Figure S4: The enriched GO terms for the contracted gene families of *E. moorcroftii*; Figure S5: Venn diagram showing the shared and specific gene families among *E. moorcroftii*, *E. mollis*, *H. rhamnoides,* and *Z. jujube*; Figure S6: The enriched GO terms for the unique gene families of *E. moorcroftii*; Figure S7: Syntenic dot plot of the self-comparison of *E. moorcroftii*; Table S1: Information of plant genomes used in this study; Table S2: Basic statistics of PacBio's HiFi sequencing data; Table S3: Mapping rates of RNA-seq reads onto the assembly genome; Table S4: Genome assembly completeness evaluated based on BUSCO; Table S5: Information of different classes of repetitive sequences in *E. moorcroftii* genome; Table S6: Functional annotation of the predicted genes in *E. moorcroftii*; Table S7: *E. moorcroftii* genome annotation completeness evaluated based on BUSCO; Table S8: List of non-coding RNA genes in *E. moorcroftii* genome.

**Author Contributions:** Conceptualization, X.F., S.W. and K.S.; methodology, J.W., X.M. and K.L.; software, J.W. and X.M.; validation, X.F., S.W. and K.S.; formal analysis, J.W. and X.F.; investigation, X.F.; resources, S.W.; data curation, J.W.; writing—original draft preparation, X.F.; writing—review and editing, S.W., K.S. and H.Z.; visualization, X.F. and J.W.; supervision, K.S.; project administration, X.F.; funding acquisition, X.F. and K.S. All authors have read and agreed to the published version of the manuscript.

**Funding:** This research was partly funded by the Shenzhen Key Laboratory of Southern Subtropical Plant Diversity, Fairy Lake Botanical Garden (Grant No. SSTLAB-2022-01 to X.F.), 2022 Key Talent Project of Gansu Province (to K.S.), and the Fundamental Research Funds for the Central Universities (lzujbky-2021-49) (to S.W.).

**Institutional Review Board Statement:** Not applicable.

**Data Availability Statement:** All raw sequencing data and the assembled genome for this project were deposited in the National Genomics Data Center (NGDC; https://ngdc.cncb.ac.cn/ (accessed on 29 July 2021)) under accession number CRA007134.

**Acknowledgments:** We would like to thank the staff in Minqin Desert Botanical Garden for their help with the plant materials' collection.

**Conflicts of Interest:** The authors declare no conflict of interest.

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
