# Peer review of "The Chromosome-Level Genome of Elaeagnus moorcroftii Wall., an Economically and Ecologically Important Tree Species in Drylands"

_diversity, doi:10.3390/d14060468_

Round 1

Reviewer 1 Report

Overall, this sequenced genome is high quality in genome assembly and annotation that not common seen in current plant genome study (possibly also associated with its small genome size of only about 500 MB). The genomic analyses were also done following a seemly  standard analysis process so that no large defect can be detected but an obvious flaw is the evolutionary analyses (see below):

       In the part for 'Evolutionary history of E. moorcroftii ', the authors reconstructed a phylogenetic tree including 11 species and further inferred the divergence time for these species/clades, but a obvious flaw is the selection for the fossils that used for time calibration:

     1): 'the root age was set to less than 125 Ma due to the fossil appearance of tricolpate pollen  in eudicots ',  please note,  125 Ma is smallest age rather than oldest age.

    2) I note a recent study that report the earliest fossil age in Nature,  by discovery of 100 Ma old fossil in the extant genus Phylica of Rhamnaceae, so this time is quite useful for time calibration of family Rhamnaceae by set the divergence time for splitting of genus Ziziphus with other species not less than 100 Ma!!----please note the current figure 2 calculated a divergence time for Ziziphus with other species by only '83' Ma !! So this divergence time for the whole tree is wrong!  

       Reference:  Shi, C., Wang, S., Cai, H. H., Zhang, H. R., Long, X. X., Tihelka, E., ... & Spicer, R. A. (2022). Fire-prone Rhamnaceae with South African affinities in Cretaceous Myanmar amber. Nature Plants, 8(2), 125-135.    

3)''Evolutionary genomic analysis 20 showed that E. moorcroftii split with E. mollis at 6.16 million years ago (Ma)'', based on the question above, this time is lower estimated.

Author Response

Response to Reviewer 1 Comments

The authors provided a high-quality genome of Elaeagnus moorcroftii Wall. Through comparative genomic analysis, they gained some new insights from the genome evolution, gene family expansion and contraction, and the inference of whole genome duplication events. This is a valuable and meaningful research. I have several concerns as follows:

We are highly grateful for your positive comments.

Point 1: Line 184: Why are collinear blocks connected by different color curves?

Response 1: The collinear blocks of different pesudochromosomes were connected by different color curves, and these too many color curves might be difficult to distinguish and further caused confusion. In this revised version, the intragenomic collinear blocks were showed in a new supplementary Figure S2, and here replaced with a photo of fruits of E. moorcroftii (Figure 1b).

Point 2: Line 299: E. moorcroftii and E. mollis are two species with close evolutionary relationship, but the family contraction and expansion patterns in these two genomes are opposite, and there are great differences, which is a very interesting discovery. I think there must be some gene families that expand in E. moorcroftii but show contraction patterns in E. mollis. For such families, more detailed comparisons should be made to reveal the possible reasons for the different evolutionary patterns of the family, or the living environment, or other factors.

Response 2: Good suggestion. We have identifed different modes of gene duplications related to expanded and contrased gene families, to explore the possible reasons for the different evolutionary patterns of gene families in two close related species. Actually, we found that dispersed duplication-derived genes contributed to more in E. mollis than in E. moorcroftii related to the expanded gene families (16.45% versus 7.64%). Tandem duplicated genes contributed to higher percentage of expanded genes in E. moorcroftii than in E. mollis (14.34% versus 6.40%) (see Figure S2). Therefore, this difference might be partly resulted from different levels of contribution from various modes of gene duplications.

Point 3: Line 247: For “whole genome duplication event in E. moorcroftii”, further evidence is needed to clearly show the two WGD events occurred in the genome of E. moorcroftii. The Ks distribution in Figure 3, it seems that there is only a recent Ks peak after the core eudicot common hexaploidization event (gamma event). In addition, the result of the orthologous syntenic ratios of 1:4 can only show that the genome has duplication event(s). It is difficult to confirm whether it is one or two WGD events.

Response 3: After carefully see the Ks curves in Figure 3a, actually, it can be found two clear peaks, but not one peak, for the Ks distribution of paralogous gene pairs. Therefore, this suggest two recent WGD events might occurred in E. moorcroftii genome. According to the reviewer’s suggestion, we have added the intragenomic synteny analysis, and the dotplot showed one syntenic block corresponding to three homologous syntenic blocks (Figure S7), again supporting two relatively recent WGD events occurred in E. moorcroftii.

Point 4: Line 283: In section of “Differential evolutionary dynamics of gene families between E. moorcroftii and E. mollis”, I would like to see this part discuss what factors may have led to the contraction and expansion patterns.

Response 4: We have added more discussions on the possible factors that led to the differential contraction and expansion patterns (also see response 2).

Point 5: Line 300: If two WGD events in E. moorcroftii are shared with two closely related species (E. mollis, and Hippophae rhamnoides), it is suggested to add the Ks distribution(s) related to the species divergence to support this conclusion (or the current inference).

Response 5: Added as suggested (see the revised Figure 3a). Meanwhile, we have added related descriptions in the text. 

Reviewer 2 Report

The authors provided a high-quality genome of Elaeagnus moorcroftii Wall. Through comparative genomic analysis, they gained some new insights from the genome evolution, gene family expansion and contraction, and the inference of whole genome duplication events. This is a valuable and meaningful research. I have several concerns as follows:

1. Line 184: Why are collinear blocks connected by different color curves?

2. Line 299: E. moorcroftii and E. mollis are two species with close evolutionary relationship, but the family contraction and expansion patterns in these two genomes are opposite, and there are great differences, which is a very interesting discovery. I think there must be some gene families that expand in E. moorcroftii but show contraction patterns in E. mollis. For such families, more detailed comparisons should be made to reveal the possible reasons for the different evolutionary patterns of the family, or the living environment, or other factors.

3. Line 247: For “whole genome duplication event in E. moorcroftii”, further evidence is needed to clearly show the two WGD events occurred in the genome of E. moorcroftii. The Ks distribution in Figure 3, it seems that there is only a recent Ks peak after the core eudicot common hexaploidization event (gamma event). In addition, the result of the orthologous syntenic ratios of 1:4 can only show that the genome has duplication event(s). It is difficult to confirm whether it is one or two WGD events.

4. Line 283: In section of “Differential evolutionary dynamics of gene families between E. moorcroftii and E. mollis”, I would like to see this part discuss what factors may have led to the contraction and expansion patterns.

5. Line 300: If two WGD events in E. moorcroftii are shared with two closely related species (E. mollis, and Hippophae rhamnoides), it is suggested to add the Ks distribution(s) related to the species divergence to support this conclusion (or the current inference).

Author Response

Response to Reviewer 2 Comments

The ivestigation of Xinxing Fu, Jingjing Wu, Xiaohui Ma, Kunpeng Li, Hui Zhang, Shengdan Wu 2, and Kun Sun “The chromosome-level genome of Elaeagnus moorcroftii Wall., an economically and ecologically important tree species in dry-lands” is important for solving modern aspects of environmental problems.

We are highly grateful for your positive comments.

Point 1: The work provides new data on the basis of which it is possible to continue resolving problems with desertification and land degradation not only in China, but in the world.

Response 1: Thank you again for the positive comments. We are in full agareement with the reviewer that the high-quality genome for Elaeagnus moorcroftii provided here is helpful to resolve problems with desertification and land degradation not only in China, but in the world.

Point 2: The reviewer recommends in the “Introduction” and “Conclusion” sections, more clearly indicate the possibility of using the obtained data.

Response 2: Good suggestion. The original sequencing data and assembled genome were submited to public database and are free access to all scientists. We have indicated this in the “Data Availability Statement” section.

Point 3: Note! The full name of the species Elaeagnus moorcroftii is given when first mentioned, then given in abbreviated form.

Response 3: We have carefully chenked the abbreviated form of species names in the whole manuscript and corrected some inappropriate forms.

Point 4: The authors mentioned that the study provides new information about the species Elaeagnus moorcroftii, which, due to its natural properties, may be useful in solving environmental problems associated with desertification. But it would be good to mention that among the investigated material or the material that will be investigated in the future, it will be possible to select samples with certain properties (resistant to abiotic factors). Such material can be used either directly to solve environmental problems, or for the breeding the new ones (samples, varieties, species) to solve them.

Response 4: According to the reviewer’s suggestion, we have added more sentences “In the further, by using the E. moorcroftii genome as reference, more studies should be encouraged to investigate the unique biology of this species. For example, how could this species respond to long-term drought and salt stresses? Moreover, whole-genome resequencing analysis of samples with certain properties (such as resistant to abiotic factors) is helpful to screen better germplasm resources and breeding new varieties.” in the conclusion part.

Point 5: Comments are advisory in nature.

Response 5: All your suggestions are helpful to improve the manuscript, and we have carefully considered all of them.

Reviewer 3 Report

The ivestigation of Xinxing Fu, Jingjing Wu, Xiaohui Ma, Kunpeng Li, Hui Zhang, Shengdan Wu 2, and Kun Sun “The chromosome-level genome of Elaeagnus moorcroftii Wall., an economically and ecologically important tree species in dry-lands” is important for solving modern aspects of environmental problems.

The work provides new data on the basis of which it is possible to continue resolving problems with desertification and land degradation not only in China, but in the world.

The reviewer recommends in the “Introduction” and “Conclusion” sections, more clearly indicate the possibility of using the obtained data.

Note! The full name of the species Elaeagnus moorcroftii is given when first mentioned, then given in abbreviated form.

The authors mentioned that the study provides new information about the species Elaeagnus moorcroftii, which, due to its natural properties, may be useful in solving environmental problems associated with desertification. But it would be good to mention that among the investigated material or the material that will be investigated in the future, it will be possible to select samples with certain properties (resistant to abiotic factors). Such material can be used either directly to solve environmental problems, or for the breeding the new ones (samples, varieties, species) to solve them.

Comments are advisory in nature.

Author Response

Response to Reviewer 3 Comments

Overall, this sequenced genome is high quality in genome assembly and annotation that not common seen in current plant genome study (possibly also associated with its small genome size of only about 500 MB). The genomic analyses were also done following a seemly standard analysis process so that no large defect can be detected but an obvious flaw is the evolutionary analyses (see below):

We thank the reviewer for the positive comments.

In the part for 'Evolutionary history of E. moorcroftii ', the authors reconstructed a phylogenetic tree including 11 species and further inferred the divergence time for these species/clades, but a obvious flaw is the selection for the fossils that used for time calibration:

1): 'the root age was set to less than 125 Ma due to the fossil appearance of tricolpate pollen  in eudicots ',  please note,  125 Ma is smallest age rather than oldest age.

Response 1: Sorry for the mistake. In this revised version, we have removed this calibration point in the dating analysis, but instead added the Phylica fossil based on your suggestion.

2) I note a recent study that report the earliest fossil age in Nature,  by discovery of 100 Ma old fossil in the extant genus Phylica of Rhamnaceae, so this time is quite useful for time calibration of family Rhamnaceae by set the divergence time for splitting of genus Ziziphus with other species not less than 100 Ma!!----please note the current figure 2 calculated a divergence time for Ziziphus with other species by only '83' Ma !! So this divergence time for the whole tree is wrong!  

Reference:  Shi, C., Wang, S., Cai, H. H., Zhang, H. R., Long, X. X., Tihelka, E., ... & Spicer, R. A. (2022). Fire-prone Rhamnaceae with South African affinities in Cretaceous Myanmar amber. Nature Plants, 8(2), 125-135.  

Response 2: We thank the reviewer for providing this useful information. As the reviewer’s suggestion, we have added the Phylica fossil as one of the calibration points. Furthermore, the time-calibrated phylogentic tree was also updated in this revised version (see the revised Figure 2).

3)''Evolutionary genomic analysis 20 showed that E. moorcroftii split with E. mollis at 6.16 million years ago (Ma)'', based on the question above, this time is lower estimated.

Response 3: The estimated divergence time for the split between E. moorcroftii and E. mollis was estimated to be 9.38 Ma based on the newly dating analysis. We have also updated this in the text.